# Socioeconomic deprivation and regional variation in Hodgkin's lymphoma incidence in the UK: a population-based cohort study of 10 million individuals

Meena Rafiq,[1] Andrew Hayward,[2] Charlotte Warren-Gash,[3] S Denaxas,[1] Arturo Gonzalez-Izquierdo,[1] Georgios Lyratzopoulos,[4] Sara Thomas[5]

For numbered affiliations see end of article.

**Correspondence to**
Dr Meena Rafiq;
mrafiq@doctors.org.uk

## ABSTRACT

**Objectives** Hodgkin's lymphoma (HL) is the the most common cancer in teenagers and young adults. This nationwide study conducted over a 25-year period in the UK investigates variation in HL incidence by age, sex, region and deprivation to identify trends and high-risk populations for HL development.

**Design** Population-based cohort study.

**Setting** Clinical Practice Research Datalink (CPRD) electronic primary care records linked to Hospital Episode Statistics and Index of Multiple Deprivation data were used.

**Participants** Data on 10 million individuals in the UK from 1992 to 2016 were analysed.

**Primary and secondary outcome measures** Poisson models were used to explore differences in HL incidence by age, sex, region and deprivation. Age-specific HL incidence rates by sex and directly age-standardised incidence rates by region and deprivation group were calculated.

**Results** A total of 2402 new cases of HL were identified over 78 569 436 person-years. There was significant variation in HL incidence by deprivation group. Individuals living in the most affluent areas had HL incidence 60% higher than those living in the most deprived (incidence rate ratios (IRR) 1.60, 95% CI 1.40 to 1.83), with strong evidence of a marked linear trend towards increasing HL incidence with decreasing deprivation (p=<0.001). There was significant regional variation in HL incidence across the UK, which persisted after adjusting for age, sex and deprivation (IRR 0.80–1.42, p=<0.001).

**Conclusions** This study identified high-risk regions for HL development in the UK and observed a trend towards higher incidence of HL in individuals living in less deprived areas. Consistent with findings from other immune-mediated diseases, this study supports the hypothesis that an affluent childhood environment may predispose to development of immune-related neoplasms, potentially through fewer immune challenges interfering with immune maturation in early life. Understanding the mechanisms behind this immune dysfunction could inform prevention, detection and treatment of HL and other immune diseases.

### Strengths and limitations of this study

► Our population-based data covered a large representative sample of over 10 million individuals in the UK over a 25-year period with 78 million years of follow-up.

► We used UK primary care electronic health records linked to secondary care data and Index of Multiple Deprivation data to improve capture of Hodgkin's lymphoma diagnoses and allow analysis of geographical and deprivation-based trends.

► Data from the Clinical Practice Research Datalink (CPRD) used in this study have been demonstrated to be generalisable to the UK population across a number of demographics.

► Data in this study were not linked to the National Cancer Register (NCR), which is a potential limitation; however lymphoma diagnosis in CPRD has been validated in previous studies and shown to have high concordance with the NCR.

► This is a cohort study of a representative sample of the UK population and not the whole UK population.

## INTRODUCTION

Hodgkin's lymphoma (HL) is the the most common cancer in teenagers and young adults worldwide.[1 2] In the UK, 2100 new cases of HL are diagnosed each year, but little is known about the distribution of these cases in the UK population or if there are any high-risk groups. International studies have identified that HL incidence varies considerably between countries, with higher rates observed in high-income countries.[3–7] This pattern is also seen within countries, with the US studies showing higher rates in more affluent regions and geographical variation in HL incidence between different states.[8] Few UK studies have investigated HL incidence patterns by socioeconomic deprivation[9–12] and region[12–14] and the results have been conflicting and inconclusive. In addition, to our knowledge there have been no recent



studies investigating patterns of HL incidence in the UK population since 2010. Understanding how HL incidence varies between different geographical regions in the UK and identifying high-risk populations may provide clues to the underlying aetiology of the disease and inform future research directions. We aimed to conduct a population-based cohort study of 10 million individuals over a 25-year period using linked primary and secondary care electronic health records to investigate variation in HL incidence within the UK by age, sex, geographic region and deprivation.

## MATERIALS AND METHODS
### Data sources and study population
Data were obtained from the UK Clinical Practice Research Datalink (CPRD), linked to Hospital Episode Statistic (HES) inpatient data and Index of Multiple Deprivation (IMD) data. CPRD is an electronic health record database containing prospectively collected pseudo-anonymised data from UK primary care consultations. It is the largest source of longitudinal primary care data, holding information on 22 million patients representing approximately 9% of the UK population (in 2013).[15] This database has been shown to be largely representative of the UK population across a number of demographics including age, sex and ethnicity.[15] Data are available from 1987 onwards when CPRD was first established. Practices contributing to CPRD are regularly audited to ensure high-data quality and that 95% of prescribing and morbidity events are captured before practices are declared 'up-to-standard' (UTS) for research purposes.[16] HES data provide additional information from hospital attendances in England. IMD scores represent a composite ecological (small-area based) measure of the socioeconomic status of a patient, based on the income, employment, disability, educational attainment and other attributes of the Local Super Output Area (LSOA) of a patient's residence. The latter typically comprise populations between 1000 and 3000 residents. All patients had an aggregate IMD score pertaining to the LSOA of their own residence (0.1% of population) or that of their general practice (99.9%) taken from the earliest available linked IMD dataset (2004 for patient-level and 2009 for practice-level).

The study population comprised patients actively registered with a CPRD practice between January 1992 and December 2016 who did not have a pre-existing diagnosis of HL. In accordance with previous studies, eligible follow-up time in days for each patient was commenced from 1 year after the patient registered with the practice (to avoid capturing past diagnoses recorded retrospectively in the few months after new patient registration),[17] or from when CPRD classified the General Practice (GP) surgery to be UTS if this occurred later. Active follow-up ended when a patient received a diagnosis of HL, died, left a CPRD practice or at the last data collection date for participating practices, whichever occurred earlier.

### Classification of outcome and exposure
Data were obtained on HL diagnoses coded using Read codes (in CPRD) or the International Classifications of Diseases, 10th Revision codes (in HES) (online supplementary tables S1 and S2); age and date of diagnosis; area of residence by Strategic Health Authority region; deprivation using IMD quintiles; date of birth and sex.

### Statistical analysis
For each new case of HL, the year and age at diagnosis were determined and the patient was counted as an incident case for that calendar year and age group. The duration of active follow-up in CPRD for each individual in the study population was then calculated and used to calculate the total person-years at risk (PYAR) and to estimate crude HL incidence rates per 100 000 PYAR.

Age-specific HL incidence rates were calculated in 5-year age bands, first for persons and then stratified by sex. Age-standardised incidence rates were estimated by the direct method using the European standard population for each region and deprivation quintile. Poisson regression was used to model HL incidence rate ratios (IRRs) for region, deprivation, age and sex independently before adjusting for other variables. East of England was used as the reference category for region, as the region with the age-standardised incidence estimate that was closest to the national average.[18] Deprivation was initially included as a categorical variable in the regression analysis to calculate IRRs and then subsequently we assessed for a linear trend by deprivation quintile, first by estimating the linear effect of deprivation using likelihood ratio tests, and then investigating departure from linearity by comparing models in which deprivation was added as a non-linear vs a linear term. In addition, incidence rates by deprivation were examined to see if any variation persisted after adjusting for trends in region, and vice versa to see if trends in region were observed after adjusting for deprivation as a categorical variable. Adjusted models were also adjusted for age and sex.

HL has a bimodal age-specific incidence pattern with the first peak occurring between 15 and 34 years and a second peak between 70 and 84 years.[18] Previous studies have suggested HL in individuals aged <50 and >50 is likely to have different aetiological factors. We therefore performed pre-specified subgroup analyses by sex (male vs female) and age (≤50 vs >50 years). We additionally examined interactions between exposure variables in the final model, particularly given potential variation in risk of HL by age and sex[18] that is, age group by sex, age group by deprivation, age group by region, sex by deprivation and sex by region. Deprivation group was treated as a categorical variable in interaction terms. Analyses were performed using Stata V.15.

### Patient and public involvement
The development of the research question for this study and aspects of the study design, particularly the subgroup analysis of the outcome by sex and age group,

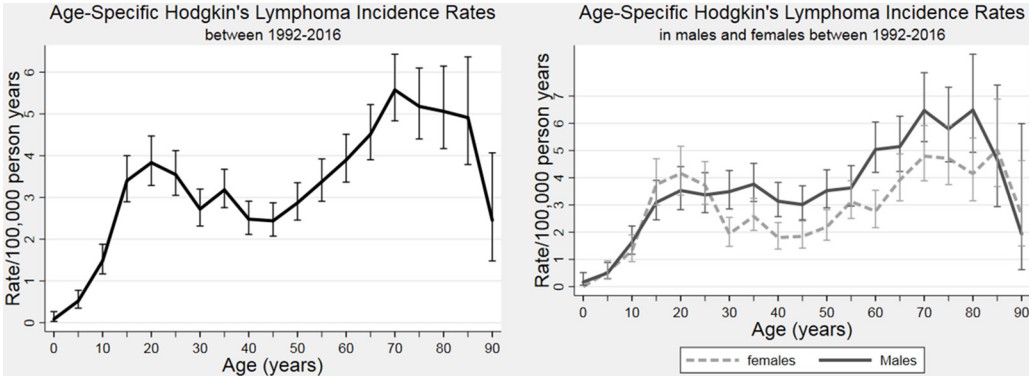

**Figure 1** Age-specific Hodgkin's lymphoma incidence in the study population (cohort of the UK population): overall (left panel) and by sex (right panel), with 95% CI bars.

were informed by discussions with HL patients' and their friends and relatives. The research focus of this study reflects their experiences and expressed research priorities in this field. Results will be shared with patient and public advisers and publicised on the CPRD website with details of the open-access paper.

## RESULTS

There were 2402 new diagnoses of HL identified over the 25-year study period (78 569 436 person-years of follow-up) with an overall HL incidence of 3.06 cases per 100 000 PYAR (95% CI 2.94 to 3.18). About 47.2% of cases were identified using CPRD alone, 16.9% were identified using HES alone and 35.9% were identified in both datasets. Age-specific HL incidence showed a bimodal distribution with an initial peak at ages 20–24 years followed by a second peak at ages 70–74 years characteristic of HL incidence in high-income countries (figure 1, online supplementary tables S3 and S4). Incidence was higher in older adults compared with those aged ≤50 (4.12 cases per 100 000 PYAR, 95% CI 3.89 to 4.36 vs 2.46 cases per 100 000 PYAR, 95% CI 2.32 to 2.59) and was higher in men than in women in all age groups (with an overall IRR for males vs females of 1.26, 95% CI 1.16 to 1.36 and age-specific IRRs ranging from 1.16 to 1.82) except for 15–29 years when incidence in female individuals exceeded that of male individuals (age-specific IRRs 0.82–0.90) and at the extremes of the age range where the number of cases were small (figure 1, online supplementary table S3). Of the 2402 incident cases of HL, 52.8% were identified in HES (407 not in CPRD), 83.1% in CPRD (1133 not in HES) and 35.9% were identified in both.

### Regional variation

Age-standardised incidence rates showed variation in HL incidence across the UK with the North East of England having the highest rates (3.89 cases per 100 000 PYAR) and Scotland having the lowest (2.35 cases per 100 000 PYAR) (figure 2, online supplementary table S4). Multivariable Poisson regression revealed strong evidence for an association between geographical region and HL incidence (table 1), which persisted after adjusting for deprivation,

age and sex (p=<0.001). Subgroup analysis showed that regional variation in HL incidence was observed in both men and women, but was limited to individuals aged over 50 years, without evidence for an association between region and HL incidence demonstrated in the younger age group (p=0.23).

### Socioeconomic deprivation

There was strong evidence for an association between HL incidence and deprivation (p≤0.001), with age-standardised incidence being highest in the most affluent population groups and lowest in the most deprived (3.92 cases per 100 000 PYAR vs 2.55 cases per 100 000 PYAR (online supplementary table S4)). Poisson regression showed that the least deprived group had HL incidence rates over 60% higher than the most deprived one, after adjusting for other factors (IRR=1.60, 95% CI 1.40 to 1.83, table 1). The strong evidence of a marked linear trend towards lower rates of HL incidence with increasing deprivation persisted after adjusting for region and was observed across both sexes and when analysing young and old adults separately (figure 3, table 1).

### Interaction analysis

Further exploring the subgroup analysis outlined above, there was no evidence that regional differences in HL incidence varied by age or sex ($P_{interaction}$=0.40 and 1.00, respectively). In addition, there was no evidence that the association between deprivation and HL risk varied by age or sex ($P_{interaction}$=0.57 and 0.39, respectively). The characteristic bimodal age-specific HL incidence pattern was observed in both men and women, and the association between age and HL incidence did not vary by sex ($P_{interaction}$=0.16).

## DISCUSSION

This is the largest study to date investigating variability in HL incidence by age, sex, deprivation and sub-national geography. It uses linked electronic primary care records over a 25-year period in a representative cohort of the UK population. We found strong evidence that lower levels of deprivation are associated with higher incidence

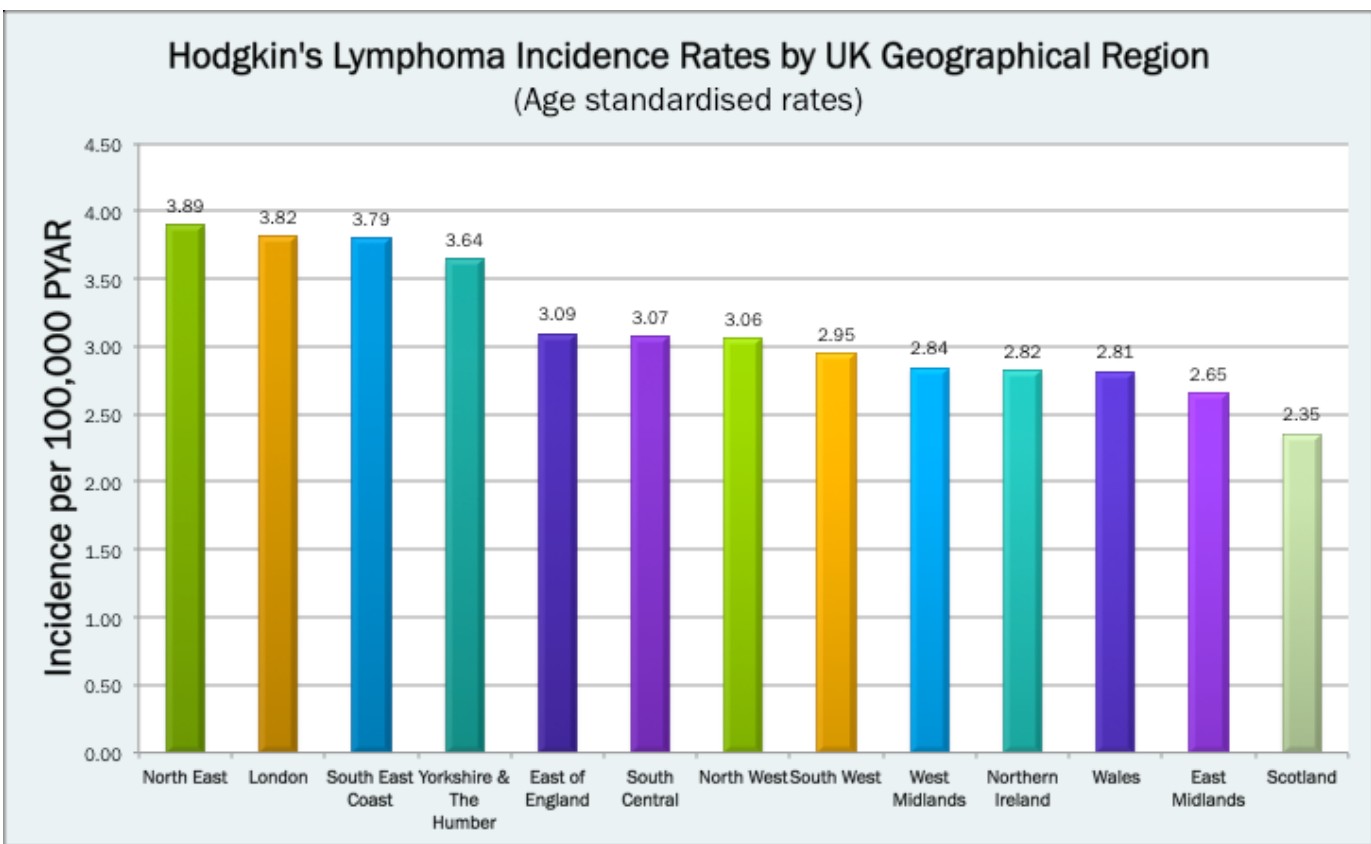

**Figure 2** Age-standardised Hodgkin's lymphoma incidence in the study population (cohort of the UK population) by region. PYAR, person-years at risk.

of HL, an association observed across age groups. There was considerable variation in HL incidence by the UK geographical region, and these differences persisted after sex, age and deprivation were taken into account.

### Comparison with the literature

The bimodal age-specific HL incidence pattern described in this study is consistent with findings from other high-income countries, including previous studies in the UK.[3 5–7 18–22] Higher incidence of HL in men except between ages 15 and 29 has also been observed in previous UK studies, which found higher incidence in women aged 15–24.[18 21] When looking at the association with deprivation previous studies have shown heterogeneous outcomes. A previous UK study found that HL incidence in men between 2006 and 2010 was greater in more deprived areas, without finding associations between deprivation and HL in the earlier study era (1996–2006) in either sex.[9] Another study investigating the distribution of childhood cancers in the UK between 1969 and 1993 found HL incidence in children aged 0–9 was greater in more deprived areas.[11] In contrast, two previous studies conducted in parts of England and Wales reported higher HL incidence with higher socioeconomic status in individuals aged 0–24,[10 12] concordant with our study findings, which are based on a population of 10 million people followed-up over a longer time period, broader geographic area and including patients other than young

adults, adolescents and children. The observation of inverse socioeconomic gradients for incidence of HL in older adults in our study is a finding which to our knowledge has not been previously reported. With regards to regional variation, the previous literature has also been conflicting. Quinn *et al* and McKinney *et al* reported no clear geographical variation in HL incidence rates within the UK population.[13 14] Alston *et al* however reported strong evidence that HL incidence varied by UK region, with greater incidence in London and the South East of England among individuals aged 0–24 years.[12] These findings concord with those of our study, which demonstrated significant regional variation in HL incidence across the UK with greater incidence in London and the South East. Regional variations were present in both men and women, but were limited to older adults, although this may reflect power limitations rather than a true lack of an effect in younger adults.

### Strengths and limitations

The main strengths of our study are that it is a large population-based study of more than 10 million individuals and has a long follow-up. HL is a relatively rare disease and the sample size and follow-up length allow for smaller effect sizes and interactions that could be missed in smaller studies to be detected. This is particularly important due to the growing evidence for two potentially separate aetiological pathways underlying HL incidence in young

**Table 1** Hodgkin's lymphoma risk by sex, socioeconomic status and geographical region

| Risk factors | Adjusted IRR (95% CI)* | | | | |
| --- | --- | --- | --- | --- | --- |
| | Study population | ≤50 Years | >50 Years | Male population | Female population |
| **Sex** | | | | | |
| Male | 1.30 (1.20 to 1.41) | 1.23 (1.10 to 1.38) | 1.38 (1.23 to 1.55) | | |
| P value | <0.001 | <0.001 | <0.001 | | |
| **Region** | | | | | |
| East of England | *ref* | *ref* | *ref* | *ref* | *ref* |
| North East England | 1.42 (1.05 to 1.93) | 0.82 (0.48 to 1.40) | 2.05 (1.40 to 3.01) | 1.49 (1.00 to 2.24) | 1.34 (0.84 to 2.13) |
| Yorkshire/Humber | 1.32 (1.04 to 1.68) | 1.11 (0.78 to 1.58) | 1.55 (1.12 to 2.13) | 1.48 (1.08 to 2.01) | 1.15 (0.79 to 1.66) |
| London | 1.29 (1.08 to 1.54) | 1.15 (0.90 to 1.48) | 1.45 (1.13 to 1.87) | 1.25 (0.99 to 1.59) | 1.33 (1.02 to 1.73) |
| South East Coast | 1.23 (1.03 to 1.48) | 1.24 (0.96 to 1.59) | 1.24 (0.97 to 1.59) | 1.19 (0.94 to 1.51) | 1.29 (0.99 to 1.69) |
| North West England | 1.07 (0.89 to 1.28) | 1.05 (0.82 to 1.36) | 1.07 (0.83 to 1.39) | 1.13 (0.89 to 1.43) | 0.99 (0.75 to 1.31) |
| South West England | 1.06 (0.87 to 1.29) | 1.08 (0.82 to 1.43) | 1.04 (0.78 to 1.37) | 1.05 0.81 to 1.36) | 1.07 (0.80 to 1.43) |
| West Midlands | 1.00 (0.82 to 1.21) | 0.90 (0.68 to 1.20) | 1.10 (0.83 to 1.44) | 0.96 (0.74 to 1.26) | 1.04 (0.78 to 1.40) |
| Wales | 0.97 (0.80 to 1.18) | 1.08 (0.83 to 1.42) | 0.87 (0.65 to 1.15) | 0.96 (0.74 to 1.25) | 0.98 (0.73 to 1.32) |
| South Central England | 0.96 (0.80 to 1.15) | 0.96 (0.75 to 1.24) | 0.95 (0.73 to 1.23) | 0.87 (0.68 to 1.11) | 1.08 (0.82 to 1.42) |
| East Midlands | 0.95 (0.73 to 1.23) | 1.10 (0.78 to 1.54) | 0.79 (0.53 to 1.17) | 0.94 (0.67 to 1.34) | 0.96 (0.65 to 1.42) |
| Northern Ireland | 0.90 (0.68 to 1.17) | 0.78 (0.53 to 1.15) | 1.02 (0.70 to 1.48) | 0.90 (0.63 to 1.29) | 0.90 (0.60 to 1.06) |
| Scotland | 0.80 (0.66 to 0.98) | 0.89 (0.68 to 1.17) | 0.71 (0.53 to 0.96) | 0.82 (0.63 to 1.08) | 0.78 (0.57 to 1.06) |
| P value | <0.001 | 0.23 | <0.001 | 0.002 | 0.03 |
| **IMD quintile** | | | | | |
| 5 (most deprived) | *ref* | *ref* | *ref* | *ref* | *ref* |
| 4 | 1.10 (0.96 to 1.26) | 1.20 (1.00 to 1.45) | 1.01 (0.83 to 1.23) | 1.11 (0.92 to 1.34) | 1.09 (0.89 to 1.33) |
| 3 | 1.15 (1.00 to 1.32) | 1.12 (0.92 to 1.36) | 1.18 (0.97 to 1.44) | 1.21 (1.00 to 1.47) | 1.08 (0.88 to 1.33) |
| 2 | 1.35 (1.18 to 1.55) | 1.37 (1.13 to 1.66) | 1.33 (1.10 to 1.62) | 1.45 (1.21 to 1.75) | 1.25 (1.02 to 1.52) |
| 1 (least deprived) | 1.60 (1.40 to 1.83) | 1.55 (1.29 to 1.88) | 1.63 (1.35 to 1.97) | 1.87 (1.57 to 2.24) | 1.31 (1.07 to 1.61) |
| P value | <0.001† | <0.001† | <0.001† | <0.001† | 0.003† |

*Adjusted IRR, Incidence rate ratio adjusted for age, sex, region and IMD quintile.
†P value from test for linear trend.
IMD, Index of Multiple Deprivation; p, p value from likelihood-ratio test; ref, reference group (East of England used as the reference category as the region with age-standardised incidence estimate that was closest to the national average).

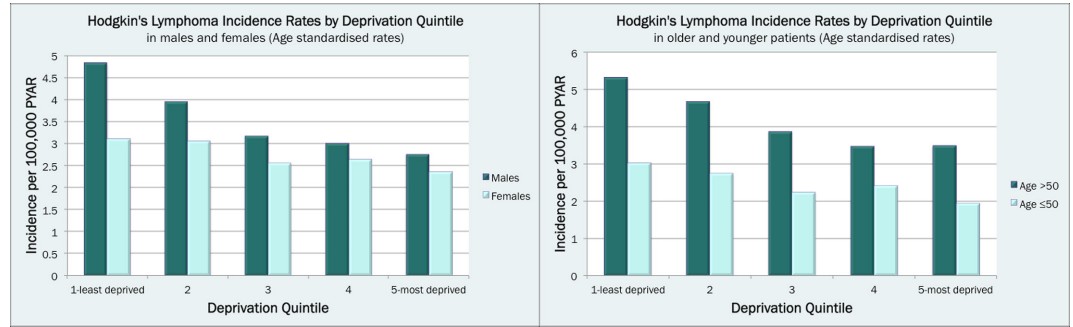

**Figure 3** Age-standardised Hodgkin's lymphoma incidence in the study population (cohort of the UK population) by deprivation: in men and women (left panel) and in individuals aged ≤50 compared with >50 (right panel). PYAR, person-years at risk.

and older adults, and therefore the need for them to be analysed independently.[3 4 23] A further advantage of this study was the use of CPRD data with regional information, linked to HES and deprivation data. CPRD has wide national coverage and has been demonstrated to be representative of the UK population across a number of demographics making the results generalisable to the UK population.[15]

The main limitation of this study is the the that it did not have access to linked data from the UK National Cancer Registry (NCR), which can be considered to represent the gold standard for estimating HL incidence. This could result in potential misclassification of cases and controls in this study and subsequent underestimation of effect estimates. Previous concordance studies have demonstrated that HL diagnoses have high validity in CPRD when compared with the NCR (positive predictive value for lymphoma 89.6%, sensitivity 97.3%) and any such effect is therefore unlikely to have materially affected the findings.[24] In addition, CPRD has established use in cancer epidemiology in the literature[24–27] and previous population-based cohort studies have demonstrated the feasibility of HL research using CPRD.[28–30] Outcome misclassification in this study was further reduced through use of HES-linked data, which improved validity of HL diagnoses by supplementing GP records with hospital data to capture cases that might have been missed in CPRD. Data was also not available on HL subtype and EBV positivity status, which would be informative for subgroup analysis to assess if trends in deprivation varied by histological group. This could be explored in future studies. Another limitation is the use of routinely collected data with potential misclassification of an individual's deprivation group. Deprivation was determined using IMD which is based on the postcode of the patients residence or registered GP practice and not on individual-level characteristics. As there may be variation in deprivation within a postcode, especially in highly diverse inner-city areas, this could result in non-differential misclassification of deprivation and underestimation of any effects. Additionally, the deprivation quintile and region captured from the dataset and used in the study may not represent childhood deprivation groups and region of residence, which may be more appropriate if early-life exposures are involved in the aetiology of HL. The earliest available linked IMD scores (2004 for patient level and 2009 for practice-level) were used in this study to estimate deprivation. This assumes both that an individuals IMD status remains stable throughout their life, and that the IMD quintile of a postcode remains stable over time. Both of these assumptions may not be true as individuals can move between deprivation quintiles and areas may undergo gentrification over time. Population movement also means an individuals childhood residence may differ from their current regional residence, which could dilute any regional variation observed in HL incidence.

## Implications

The bimodal incidence pattern and differences in regional variation between younger versus older adults supports the hypothesis that there may be different aetiological pathways involved in the development of HL in these age groups.[3 4 19] This is further supported by evidence from previous studies for different distributions of the histological subtypes of HL between the two age groups.[5 7 8 23 31–33] Consideration should be given to investigating HL aetiology separately in these age groups in future studies to identify potential different contributory factors that could be masked when analysing the population as a whole. In addition, the existence of potentially different pathophysiology could have important implications for targeting and response to treatment regimens and in disease monitoring and detection. The peak in disease incidence in young female adults is characteristic of a number of immune-related conditions, including multiple sclerosis, rheumatoid arthritis and lupus.[34–36] Similarities between incidence patterns for these diseases could suggest a common predisposing factor in early life that interferes with immune regulation and promotes development of immune-related diseases in young adults.

The trend towards increased HL incidence with increased affluence was replicated across three separate UK databases and is consistent with findings from the US studies.[8 10 12] Concordance between these findings add further support for this being a true association. This trend has been previously well established in ecological studies making comparisons between countries with very different levels of deprivation.[3–7] Within country differences in deprivation tend to be much smaller than those seen between countries. Our results could suggest that even small increases in community deprivation levels may elevate an individual's risk of developing HL. A proposed explanation for this association is that children in affluent households with less overcrowding and cleaner childhood environments consequently have delayed exposure to infectious agents and fewer immune challenges in early-life to stimulate immune development and regulation.[37–42] This predisposes them to develop immune-related conditions. This phenomenon has been demonstrated for other haematological malignancies, including leukaemia, where low-infection burden and lack of microbial exposure in early life were found to result in immune system malfunction and were associated with increased risk of developing subsequent leukaemia.[43] Observation of this trend in older adults is less likely to be explained by childhood exposures. HL aetiology could be multifactorial with childhood exposures predisposing individuals, but in the absence of other promoting factors in early-life, onset of HL is delayed until later adulthood. This should be further explored in future studies to identify contributory factors underlying the association in older adults.

Regional variation in HL incidence was observed after adjusting for deprivation differences in older adults. This indicates that other factors that vary geographically in these regions are contributing to increased HL incidence

in this age group. Geographical clustering of HL cases has been previously reported in both the UK and the USA,[8 12 44–47] which could support the role for an environmental factor underlying increased rates in these regions. Other possible contributory factors include regional differences in ethnicity and clustering of predisposing or protective genotypes. Further studies are required to investigate the role of these different factors in regional variation in UK HL incidence.

## CONCLUSION

This study of over 10 million individuals based on nationwide primary care data found strong evidence for regional variation in HL incidence across the UK that cannot be explained by geographical differences in deprivation. More affluent individuals within the UK population have a significantly higher risk of developing HL in both younger and older adults. This trend has been observed for other immune-mediated diseases. The findings are consistent with the hypothesis that an affluent childhood environment may predispose to development of immune-related conditions, possibly through fewer immune challenges interfering with the maturation of the immune system. Further understanding the responsible pathophysiological mechanisms could inform prevention, detection and treatment of HL and other immune conditions.

**Author affiliations**
[1]Institute of Health Informatics, University College London, London, UK
[2]Institute of Epidemiology and Health Care, University College London, London, UK
[3]Non-communicable Disease Epidemiology, London School of Hygiene and Tropical Medicine, London, UK
[4]Department of Behavioural Science and Health, ECHO (Epidemiology of Cancer Healthcare & Outcomes) Research Group, University College London, London, UK
[5]Non-communicable Disease Epidemiology, London School of Hygiene and Tropical Medicine, London, UK

**Acknowledgements** We thank the patient and public advisers who assisted in the design and conception of this study.

**Contributors** MR, AH, CW-G and ST contributed to the conception and design of the study, planning of analyses, interpretation of results and writing the manuscript. MR, SD and AG-I contributed to the planning of the analyses, extracting the data and performing the statistical analyses. GL contributed to study design and interpretation of results. All authors have read and approved the final manuscript.

**Funding** MR and the work presented in this paper are funded by the National Institute for Health Research (NIHR) in-practice clinical fellowship (IPF-2017-11-011). This article presents independent research funded by the NIHR. The views expressed are those of the author(s) and not necessarily those of the NHS, the NIHR or the Department of Health. GL is supported by the Cancer Research UK Advanced Clinician Scientist Fellowship (C18081/A18180).

**Competing interests** None declared.

**Patient consent for publication** Not required.

**Ethics approval** The protocol for this project was approved by the LSHTM Ethics Committee (ref:11182) and the ISAC for MHRA Database Research (protocol number:16_237). Generic ethical approval for observational studies conducted using anonymised CPRD data with approval from ISAC has been granted from a National Research Ethics Service Committee (NRESC).

**Provenance and peer review** Not commissioned; externally peer reviewed.

**Data availability statement** Data may be obtained from a third party and are not publicly available.

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
