## [Reviewer comments · BMJ Open]

ARTICLE DETAILS

TITLE (PROVISIONAL)	Socioeconomic deprivation and regional variation in Hodgkin's lymphoma incidence in the UK: A population-based cohort study of 10 million individuals
AUTHORS	Rafiq, Meena; Hayward, Andrew; Warren-Gash, Charlotte; Denaxas, S; Gonzalez-Izquierdo, Arturo; Lyratzopoulos, Georgios; Thomas, Sara

VERSION 1 – REVIEW

REVIEWER	Alex Smith University of York, UK
REVIEW RETURNED	17-Feb-2019

GENERAL COMMENTS	The authors have undertaken a study examining the incidence of Hodgkin lymphoma in newly diagnosed cases between 1992-2016 and the association with socio-economic status as measured by the area-based Index of Multiple Deprivation (IMD) with cases ascertained from Hospital Episode Statistics (HES) and Clinical Practice Research Datalink (CPRD). There are some serious methodological issues that require addressing: 1. Why use routine data sources instead of using national cancer registry data for a cancer that is well ascertained in national systems? The rationale for using CPRD to ascertain cases is not clear.2. The authors have captured all new cases of HL newly diagnosed 1992-2016, however, they only have used a measure of IMD from one time-point (2004). This is only mentioned in the Discussion and thus should be described in the Methods. As the authors themselves acknowledge the use of a measure from one time-point (2004) means that large assumptions are being made around a person's socio-economic status and assumes an area measure of SES remaining static over time, both of which isn't correct. Furthermore, the measure is assigned at the time of diagnosis, and yet the authors draw conclusions around "childhood exposures" and risk of developing HL across all ages.3. The literature quoted to support the findings of this study have generally been derived from historical studies, yet more contemporary data from national registration that does not show an association have not been quoted, for example the latest report by National Cancer Intelligence "Cancer by Deprivation in England". Based on the case ascertainment and how the measure of exposure (deprivation) has been defined means that the
---

	conclusions drawn by the authors “affluent childhood environment with fewer immune challenges affects immune maturation in early life, thereby predisposing to development of immune-related neoplasms” are not robust or supported by the findings of this study. Minor comments 4. All patients had an aggregate IMD score pertaining to the LSOA of their own residence or that of their general practice. What proportion could not be assigned a score based in their own residence? 5. Abstract - need to add CPRD as a data source along with HES in the Setting sections, as this was the main source of ascertainment.
--	---

REVIEWER	Irene Biasoli Federal University of Rio de Janeiro Associate Professor
REVIEW RETURNED	17-Feb-2019

GENERAL COMMENTS	This is a population-based data with a large representative sample of over 10 87 million individuals in the UK over a 25-year period. The authors used UK primary care electronic health records and Index of Multiple Deprivation. They found 2,402 new cases of HL, and There was a significant variation in HL incidence by deprivation group. Individuals living in the most affluent areas had HL incidence 60% higher than those living in the most deprived with strong evidence of a marked linear trend towards increasing HL incidence with decreasing deprivation. The article is interesting and goes deep in a particularly sensitive topic. Methods are accurately described methods and statistical analysis well-developed. Results and tables are well-presented. My only observation regards a specific point. It would be informative if you have the subtypes of Hodgkin Lymphoma and EBV positivity and analyse the association with a deprivation index, if these information is available in the dataset. Also, in the discussion you can add some comments about the same phenomenon related to affluent childhood environment with fewer immune challenges interferes with the maturation of the immune system and predisposes of association of incidence and socioeconomic status in other hematologic malignancies as for example childhood leukemia. These observations don't compromise the quality of the article.
--

VERSION 1 – AUTHOR RESPONSE

Reviewer 1:

Reviewer comments: Alex Smith	Response
The authors have undertaken a study examining the incidence of Hodgkin lymphoma in newly diagnosed cases between 1992-2016 and the association with socio-economic status as measured by the area-based Index of Multiple Deprivation (IMD) with cases ascertained from Hospital Episode Statistics (HES) and Clinical Practice Research Datalink (CPRD).	Thank you for aptly summarising the design and purpose of our study. We appreciate the Reviewer's expertise in the field of epidemiology of haematological cancers.
1. Why use routine data sources instead of using national cancer registry data for a cancer that is well ascertained in national systems? The rationale for using CPRD to ascertain cases is not clear.	We agree with the Reviewer that population-based national cancer registry (NCR) data are optimal for studying variation in incidence. The present study was linked to a broader project aiming to examine associations between certain types of diseases commonly encountered in primary care (treated as exposures) and Hodgkin Lymphoma (HL) (treated as an outcome); this is a research question that obligates the use of primary care records data (given that, by their design and purpose, rich information on diseases other than cancer is not typically encountered in cancer registries). Within this broader study, we use CPRD data to characterise the cohort with respect to incident variation. CPRD has very good validity as a source of data on cancer incidence, as shown in validation studies (which we cite) indicating high concordance (>90%) with cancer registration data (references 24-27). Additionally, previous studies have demonstrated the feasibility of epidemiological research on Hodgkin's lymphoma using CPRD data (references 28-30). The potential for under-recording of incident cases in our data is further reduced by linkage to HES data, which increased capture of cases. We have added an additional sentence to the first paragraph of the results section outlining this as follows: "47.2% of cases were identified using CPRD alone, 16.9% were identified using HES alone and 35.9% were identified in both datasets." We posit that any impact from HL cases not recorded in CPRD would likely be small (and further reduced through the linkage to HES). Further, any potential under-recording is unlikely to be differential (by age, sex, geographical region). The issue is alluded to, and additional information/references presented, in the Limitations part of Discussion (paragraph 4,

	Limitations subparagraph an references 24-30 in the manuscript). References: 24. Boggon R, van Staa TP, Chapman M, Gallagher AM, Hammad TA, Richards MA. Cancer recording and mortality in the General Practice Research Database and linked cancer registries. Pharmacoepidemiol Drug Saf. 2013;22(2):168-75. 25. Dregan A, Moller H, Murray-Thomas T, Gulliford MC. Validity of cancer diagnosis in a primary care database compared with linked cancer registrations in England. Population-based cohort study. Cancer Epidemiol. 2012;36(5):425-9. 26. Arhi CS, Bottle A, Burns EM, Clarke JM, Aylin P, Ziprin P, et al. Comparison of cancer diagnosis recording between the Clinical Practice Research Datalink, Cancer Registry and Hospital Episodes Statistics. Cancer Epidemiol. 2018;57:148-57. 27. Margulis AV, Fortuny J, Kaye JA, Calingaert B, Reynolds M, Plana E, et al. Validation of Cancer Cases Using Primary Care, Cancer Registry, and Hospitalization Data in the United Kingdom. Epidemiology (Cambridge, Mass). 2018;29(2):308. 28. Strongman H, Brown A, Smeeth L, Bhaskaran K. Body mass index and Hodgkin's lymphoma: UK population-based cohort study of 5.8 million individuals. British journal of cancer. 2019:1. 29. Castellsague J, Kuiper JG, Pottegård A, Berglind IA, Dedman D, Gutierrez L, et al. A cohort study on the risk of lymphoma and skin cancer in users of topical tacrolimus, pimecrolimus, and corticosteroids (Joint European Longitudinal Lymphoma and Skin Cancer Evaluation-JOELLE study). Clinical epidemiology. 2018;10:299. 30. Gelfand JM, Berlin J, Van Voorhees A, Margolis DJ. Lymphoma rates are low but increased in patients with psoriasis: results from a population-based cohort study in the United Kingdom. Archives of dermatology. 2003;139(11):1425-9.
2. The authors have captured all new cases of HL newly diagnosed 1992-2016, however, they only have used a measure of IMD from one time-point (2004). This is only mentioned in the Discussion and thus should be described in the Methods. As the authors themselves acknowledge the use of a measure from one time-point (2004) means that large assumptions are	We agree with the comment on describing the time point of IMD measurement in Methods, and have now done so Methods (See Methods, end of paragraph 1). We continue to reflect on the matter in Discussion nonetheless as a limitation. We agree with the Reviewer that optimally we would have liked to have a measure of IMD early in the life-course of incident cases and controls. Unfortunately, to reduce risk of patient identification, CPRD only

being made around a person’s socio-economic status and assumes an area measure of SES remaining static over time, both of which isn’t correct. Furthermore, the measure is assigned at the time of diagnosis, and yet the authors draw conclusions around “childhood exposures” and risk of developing HL across all ages.	provide access to IMD quintiles for a single time point in any dataset that they release for research. We therefore selected the earliest available time point of IMD measurement, which is most likely to be representative of the childhood socioeconomic status. Given general social mobility patterns during life-course, we do believe that our ‘once only’ measurement of IMD is a good surrogate marker of early-life IMD quintile in most cases, though with a degree of misclassification, which may be greater for older cases. We now acknowledge this matter in the second half of paragraph 4 in the Discussion (Limitations subparagraph) as following: “Additionally, the deprivation quintile and region captured from the dataset and used in the study may not represent childhood deprivation groups and region of residence, which may be more appropriate if early-life exposures are involved in the etiology of HL. The earliest available linked IMD scores (2004 for patient level and 2009 for practice-level) were used in this study to estimate deprivation. This assumes both that an individuals IMD status remains stable throughout their life, and that the IMD quintile of a postcode remains stable over time. Both of these assumptions may not be true as individuals can move between deprivation quintiles and areas may undergo gentrification over time. Population movement also means an individual’s childhood residence may differ from their current regional residence, which could dilute any regional variation observed in HL incidence.”
3. The literature quoted to support the findings of this study have generally been derived from historical studies, yet more contemporary data from national registration that does not show an association have not been quoted, for example the latest report by National Cancer Intelligence “Cancer by Deprivation in England”. Based on the case ascertainment and how the measure of exposure (deprivation) has been defined means that the conclusions drawn by the authors “affluent childhood environment with fewer immune challenges affects immune maturation in early life, thereby predisposing to development of	We agree and reference this report in both the introduction and discussion sections (reference 9 in the manuscript). Specifically, in Discussion, ‘Comparison with other literature’, the findings of this study are described as follows “A previous UK study found that HL incidence in males between 2006-2010 was greater in more deprived areas, without finding associations between deprivation and HL in the earlier study era (1996-2006) in either sex”. We have also taken the opportunity to update the reference list and have added additional findings from population-based incidence studies in addition to reference 9 (i.e. reference 22 in the manuscript). We thank the Reviewer for highlighting that our concluding sentence was far too affirmative given the evidence in our study. We have therefore moderated the sentence in question in the Conclusion and

immune-related neoplasms” are not robust or supported by the findings of this study.	Abstract (now denoting that this is a hypothesis with which our findings are consistent with, rather than asserting a ‘proof’ as in our original) to now read: “The findings are consistent with the hypothesis that an affluent childhood environment may predispose to development of immune-related conditions, possibly through fewer immune challenges interfering with the maturation of the immune system.” We have addressed the issues of use of CPRD data for cancer incidence and the use of an ‘one-point-in-time’ IMD measurement above (see replies to comments 1 and 2). References: 9. Network NCI. Cancer by Deprivation in England. Incidence, 1996-2010. Mortality, 1997-2011. 2014. 22. Smith A, Crouch S, Lax S, Li J, Painter D, Howell D, et al. Lymphoma incidence, survival and prevalence 2004–2014: sub-type analyses from the UK’s Haematological Malignancy Research Network. British journal of cancer. 2015;112(9):1575.
4. All patients had an aggregate IMD score pertaining to the LSOA of their own residence or that of their general practice. What proportion could not be assigned a score based in their own residence?	Predominantly in our study population the aggregate IMD score relates to the residence of the general practice (99.9% versus 0.1% based on patient residence). This is because of information governance rules applied by ISAC/CPRD related to patient anonymisation and the protocol of the study. This information has now been stipulated in the Methods (end of paragraph 1 in methods).
5. Abstract - need to add CPRD as a data source along with HES in the Setting sections, as this was the main source of ascertainment.	Many thanks for bringing this to our attention. This has now been added to the abstract.

Reviewer 2:

Reviewer comments: Irene Biasoli	Response
This is a population-based data with a large representative sample of over 10 million individuals in the UK over a 25-year period. The authors used UK primary care electronic health records and Index of Multiple Deprivation. They found 2,402 new cases of HL, and There was a significant variation in HL incidence by deprivation group. Individuals living in the most affluent	Thank you for the positive appraisal and aptly summarising our paper.

areas had HL incidence 60% higher than those living in the most deprived with strong evidence of a marked linear trend towards increasing HL incidence with decreasing deprivation. The article is interesting and goes deep in a particularly sensitive topic. Methods are accurately described methods and statistical analysis well-developed. Results and tables are well-presented.	
My only observation regards a specific point. It would be informative if you have the subtypes of Hodgkin Lymphoma and EBV positivity and analyse the association with a deprivation index, if these information is available in the dataset. Also, in the discussion you can add some comments about the same phenomenon related to affluent childhood environment with fewer immune challenges interferes with the maturation of the immune system and predisposes of association of incidence and socioeconomic status in other hematologic malignancies as for example childhood leukemia. These observations don't compromise the quality of the article.	Many thanks for your useful comments. We agree that it would be informative to do a subgroup analysis by EBV status in cases. Unfortunately this information was not available within our data (and we are not aware of large population-based data sources covering a similar study period with this information). We have added this as a suggestion for future research in the discussion section as follows: “Data was also not available on HL subtype and EBV positivity status, which would be informative for subgroup analysis to assess if trends in deprivation varied by histological group. This could be explored in future studies” We agree with the Reviewer’s point regarding the same phenomenon being observed in leukaemia and other childhood malignancies. This has now been included in the discussion section with appropriate references (reference 43) as follows: “This phenomenon has been demonstrated for other haematological malignancies, including leukaemia, where low infection burden and lack of microbial exposure in early life were found to result in immune system malfunction and were associated with increased risk of developing subsequent leukaemia”. References: 43. Greaves M. A causal mechanism for childhood acute lymphoblastic leukaemia. Nature Reviews Cancer. 2018:1. The authors also provided a marked copy with additional response. Please contact the publisher for full details.

VERSION 2 – REVIEW

REVIEWER	Alexandra Smith University of York
REVIEW RETURNED	12-May-2019

GENERAL COMMENTS	Many thanks to the authors for addressing the queries around the issues of the use of CPRD data to address the question of incidence and Hodgkin lymphoma. Sorry to remain negative, but I am still not convinced that CPRD data is the optimal data source to examine regional variation in incidence, especially as CPRD coverage is not national and there are differences by region in the rate of participation by GP practices.
---

REVIEWER	irene biasoli Federal University of Rio de Janeiro, Brazil
REVIEW RETURNED	23-Apr-2019

GENERAL COMMENTS	The reviewer completed the checklist but made no further comments.
--